# Quantifying the Economic and Financial Viability of NB-IoT and LoRaWAN Technologies: A Comprehensive Life Cycle Cost Analysis Using Pragmatic Computational Tools

**Bernhard Koelmel** [1,2,3,*] **, Max Borsch** [4] **, Rebecca Bulander** [1] **, Lukas Waidelich** [1] **, Tanja Brugger** [1] **, Ansgar Kuehn** [1] **, Matthias Weyer** [1] **, Luc Schmerber** [5] **and Michael Krutwig** [6]

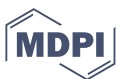

1   School of Engineering, Industrial Engineering, Pforzheim University, 75175 Pforzheim, Germany; rebecca.bulander@hs-pforzheim.de (R.B.); lukas.waidelich@hs-pforzheim.de (L.W.); tanja.brugger@hs-pforzheim.de (T.B.); ansgar.kuehn@hs-pforzheim.de (A.K.); matthias.weyer@hs-pforzheim.de (M.W.)
2   ISM International School of Management, Chair Technology Management, 75007 Paris, France
3   McCoy College of Business, Texas State University, San Marcos, TX 78666, USA
4   Wirtschaftsförderung Nordschwarzwald, 75172 Pforzheim, Germany; max.borsch@nordschwarzwald.de
5   Luc Schmerber, 76199 Karlsruhe, Germany; luc@lucschmerber.com
6   Krumedia GmbH, 76199 Karlsruhe, Germany; michael.krutwig@krumedia.com
*   Correspondence: bernhard.koelmel@hs-pforzheim.de or bkoelmel@txstate.edu

**Abstract:** This paper focuses on quantifying the economic and financial viability of NB-IoT and LoRaWAN technologies, two low-power wide-area network (LPWAN) technologies with unique characteristics that make them suitable for IoT applications. The purpose of this study is to propose a "pragmatic" artifact for performing life cycle cost analysis and demonstrate its application to these technologies. The methodology uses pragmatic computational tools to facilitate the analysis and considers all relevant economic and financial factors, such as operating costs, equipment costs, and revenue potential. The main finding of this study is that Narrow Band-Internet of Things (NB-IoT) and Long Range Wide Area Network (LoRaWAN) technologies have different cost structures and revenue potentials, which may affect their economic and financial viability for different IoT applications. Ultimately, the study concludes that a comprehensive life cycle cost analysis is critical to making informed decisions about technology adoption, and that the proposed methodology can be applied to other IoT technologies to gain insight into their economic and financial viability.

**Keywords:** financial viability; life cycle cost analysis; LPWAN; pragmatic computational tools; design science research; data-driven decision making

## 1. Introduction

The economic and financial viability of emerging technologies plays a crucial role in the decision-making process for their adoption and implementation. This scientific paper aims to explore the economic and financial viability of Narrow Band-Internet of Things (NB-IoT) and Long Range Wide Area Network (LoRaWAN) technologies for different Internet of Things (IoT) applications within I4.0/I5.0 and Smart City environments. The research question driving this study is: "What is the economic and financial viability of NB-IoT and LoRaWAN technologies for different IoT applications, as assessed through a comprehensive life cycle cost analysis?"

Our hypothesis posits that the economic and financial viability of NB-IoT and Lo-RaWAN technologies varies based on the specific IoT applications, owing to differences in cost structures and revenue potentials. To test this hypothesis, a comprehensive life cycle cost analysis will be conducted with the help of design science research, taking into account various economic and financial factors, such as operating costs, equipment costs, and revenue potential.

The findings of this research will provide valuable insights into the total cost of ownership of NB-IoT and LoRaWAN technologies, identifying potential areas for cost savings, and facilitating informed decision-making processes regarding their adoption and implementation in diverse IoT applications. Industry 4.0 and the emerging concept of Industry 5.0 have ushered in a new era of technological advancements that are reshaping industries and societies around the world [1]. These transformative technologies, collectively referred to as I4.0/I5.0 technologies, encompass a wide range of digital innovations such as the Internet of Things (IoT), cloud computing, artificial intelligence (AI), robotics, and automation. These technologies hold great promise for revolutionizing various sectors, including manufacturing, healthcare, transportation, and smart city/urban development.

The benefits of I4.0/I5.0 technologies are significant and varied. They enable increased efficiency, productivity, and quality in manufacturing processes, resulting in cost reductions, improved product customization, and faster time to market. In an economic context, these technologies form the backbone of improvement initiatives, enabling stakeholders to optimize resource utilization, improve service delivery, etc.

The adoption of I4.0/I5.0 technologies brings forth socio-economic challenges, including the digital divide, inequality of access, and workforce transformation. Interoperability presents a challenge that necessitates the establishment of standards and protocols for seamless communication and integration among diverse technologies and stakeholders [2]. Privacy and data security are also paramount concerns in the adoption of these technologies. The vast amount of data generated and exchanged raises issues of privacy, data ownership, and cybersecurity. Protecting privacy and implementing robust security measures are essential for building trust.

The adoption of I4.0/I5.0 technologies presents unique challenges, as noted above, but also in terms of financial viability [3], including lifecycle costs that include both upfront investments and operational expenses. This article focuses on evaluating the economic and financial feasibility of NB-IoT and LoRaWAN technologies in light of the challenges associated with the adoption of emerging technologies. This study aims to provide valuable insights into the viability of these technologies by assessing their potential benefits and examining the associated costs. Understanding the economic and financial aspects is crucial for organizations and stakeholders to effectively leverage NB-IoT and LoRaWAN, considering resource allocation and investments. These costs include the cost of implementation, reskilling, and training the workforce to adapt to the new technologies, and developing the infrastructure to support their integration. The lifecycle costs of communication technologies, in particular, are an important area, as they tie up capital over a long period of time. Financial analysis, including life cycle costing, is essential to ensure a comprehensive assessment of the costs and benefits associated with the adoption of a new technology [4]. This paper aims to quantify the economic and financial viability of two promising IoT technologies, NB-IoT and LoRaWAN [5], through a comprehensive life cycle cost analysis using pragmatic computational tools.

The life cycle cost analysis assesses the full range of costs and benefits associated with the deployment of NB-IoT and LoRaWAN technologies, including not only the upfront costs but also the costs associated with operations, maintenance, and disposal [6]. The analysis will provide decision-makers with an understanding of the total cost of ownership of these technologies and identify potential areas for cost savings [7].

Furthermore, this paper argues that financial analysis should accompany technology decisions to ensure that both aspects are addressed for the successful adoption of innovative technologies. In the context of the "not-invented-here" syndrome [8], which hinders good decisions about innovative technologies, a thorough financial analysis becomes critical. The syndrome can lead to the adoption of innovative technologies without proper financial analysis, resulting in inefficient use of resources, high costs, and ultimately, technology adoption failure.

Therefore, this paper proposes a methodology for conducting a life cycle cost analysis of NB-IoT and LoRaWAN technologies to quantify their economic and financial viability [9].

The methodology uses pragmatic computational tools to facilitate the analysis and ensure that it is comprehensive and efficient.

In conclusion, this paper argues that financial analysis is essential for technology adoption decisions, and a comprehensive life cycle cost analysis can facilitate the decision-making process. Furthermore, the proposed methodology can be applied to other IoT technologies to provide valuable insights into their economic and financial viability. Ultimately, this can enable organizations to make informed technology adoption decisions, maximize the benefits of innovative technologies, and minimize financial risks [10].

## 2. Design Science Research as Scientific Approach

As a research method we employ design science research in developing the artifact "Life Cycle Cost Analysis Using Pragmatic Computational Tools." Design science research as depicted in Figure 1 is a research paradigm that aims to produce innovative solutions to practical problems through the creation of new artifacts, such as models, methods, and tools [11–14]. The relevance cycle in Design Science Research ensures that the designed artifacts address real-world problems, while the rigor cycle ensures the scientific validity and quality of the research process and outcomes. The process involves identifying a problem, developing a solution, and evaluating its effectiveness. The use of design science research in information systems is increasingly popular due to its ability to produce practical and relevant solutions that can be implemented in real-world settings [15].

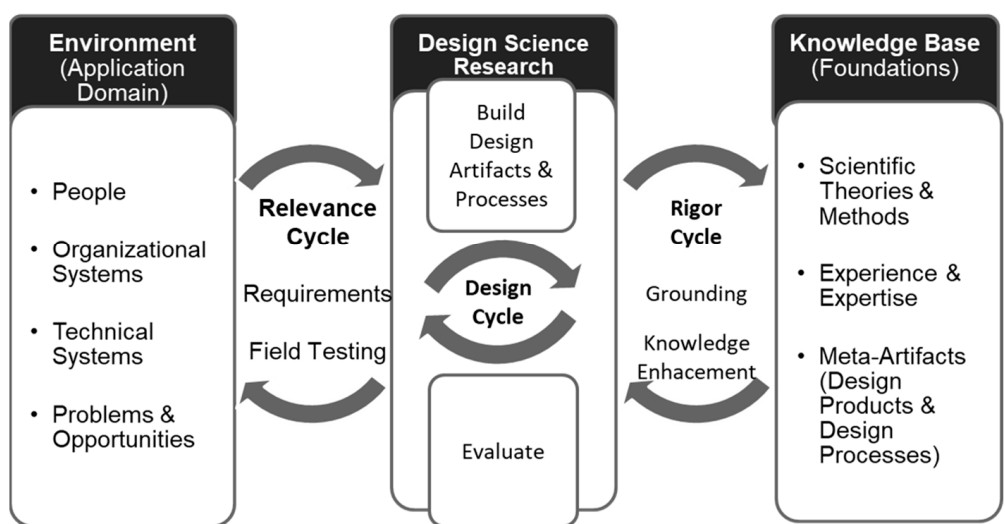

**Figure 1.** Research Methodology: Design science research; Own illustration, based on [14].

In this context, our artifact seeks to fill a significant research gap by providing a comprehensive life cycle cost analysis tool for IoT technologies such as NB-IoT and LoRaWAN.

Design Science Research (DSR) or design-oriented research is a scientific method that aims to develop practice-oriented solutions to problems or challenges. Design Science Research (DSR) holds significant practical relevance in addressing real-world problems and driving innovation in various domains. Unlike traditional research methods that focus primarily on observing and explaining phenomena, DSR emphasizes the creation of artifacts or solutions that can directly address practical challenges. This practical orientation is crucial, as it allows researchers to bridge the gap between theory and practice, translating theoretical knowledge into tangible outcomes that can be implemented and evaluated. It aims to create new knowledge by developing artifacts, such as models, methods, and tools, that can be applied in real-world settings [16]. The central tenet of DSR is that the development of a novel artifact should be grounded in a problem domain and informed by an understanding of the state of the art in the relevant field. The Design Science Research (DSR) process as shown in Figure 2 encompasses several key steps to address practical problems and develop innovative artifacts [13].

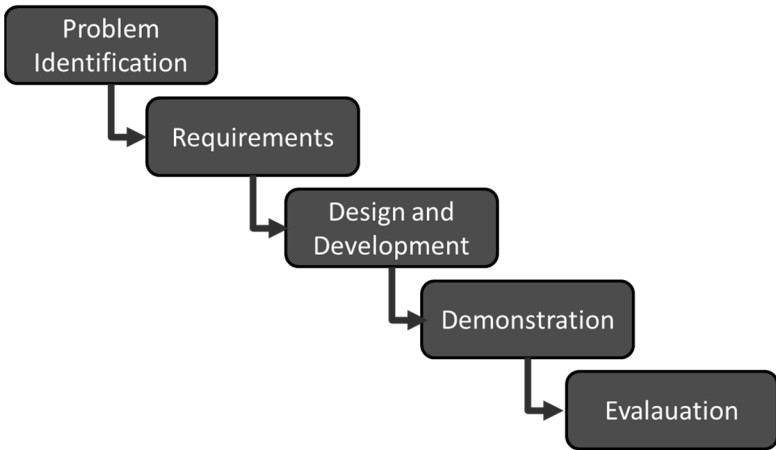

**Figure 2.** Design Science Research Process; Own illustration, based on [13].

The first step is problem identification, where researchers identify a practical problem grounded in a specific context and informed by the existing literature and practice. Once the problem is identified, the design and development phase begins, leveraging existing knowledge and theory to create an artifact that effectively addresses the identified problem. This phase may involve the creation of new theories or adapting existing theories to a new context.

The subsequent phase focuses on demonstrating the usefulness of the artifact. Researchers showcase how the artifact can be utilized to solve the practical problem identified earlier. This can be achieved through testing in simulated or real-world environments, highlighting how the artifact surpasses existing solutions or practices.

Finally, the effectiveness of the artifact is evaluated. This evaluation employs quantitative or qualitative methods to assess the impact of the artifact on the problem domain. The evaluation aims to provide evidence of the artifact's usefulness while offering insights into its limitations and potential for further development.

By following this structured process, researchers can systematically identify, design, develop, demonstrate, and evaluate artifacts, ensuring their practical applicability and advancing their knowledge in the respective field. This approach enables the generation of valuable insights into the economic, societal, and technological aspects of the artifacts, guiding decision-making processes and promoting continuous improvement.

DSR has been increasingly applied in the field of information systems and has proven to be effective in producing practical solutions to complex problems. This approach has been used to develop a wide range of artifacts, including software systems, decision support tools, and frameworks for guiding practice. DSR differs from traditional research approaches in that it places greater emphasis on the practical relevance of the research results. While traditional research may focus on developing theoretical models and testing them in a controlled setting, DSR seeks to develop solutions that can be implemented in real-world settings and have a measurable impact on practice.

## 3. State of the Art

### 3.1. IoT Communication Technologies for the Internet of Things

Low-power wide area networks (LPWANs) have become a popular communication technology for the Internet of Things (IoT) due to their low power consumption and wide coverage area. Two of the most popular LPWAN technologies are LoRaWAN and NB-IoT [17–19].

LoRaWAN (Long Range Wide Area Network) is a wireless communication protocol based on the LoRa modulation technique. LoRaWAN has the ability to communicate over long distances, typically up to 10 km in rural areas and up to 2 km in urban areas. It operates in unlicensed frequency bands, which makes it a cost-effective solution. The LoRaWAN

protocol is open-source and has a large community of developers. LoRaWAN is primarily used for battery-powered devices that require low data rates, such as environmental monitoring sensors, smart parking systems, and asset tracking [20,21].

NB-IoT (Narrowband IoT) is a cellular technology designed for IoT devices within 5G cellular networks. It is a standardization effort by the 3GPP and is based on the LTE (Long-Term Evolution) technology. The NB-IoT uses a narrow bandwidth of 200 kHz and can operate in licensed or unlicensed frequency bands. The main advantage of the NB-IoT is its ability to operate in areas with weak signal strengths and in underground locations. It can also support high data rates and has a low latency, making it suitable for applications that require real-time data, such as industrial automation and smart cities [22,23]. Table 1 summarizes the technical properties of LoRaWAN and NB-IoT.

**Table 1.** Selected properties of LoRaWAN vs. NB-IoT [20–24].

| Property | LoRaWAN | NB-IoT |
|---|---|---|
| Modulation Technique | LoRa (Chirp Spread Spectrum (CSS)) | QPSK (Orthogonal Frequency-Division Multiplexing (OFDM)) |
| Frequency Range | 868 MHz, 915 MHz, and 433 MHz | 700 MHz, 800 MHz, 900 MHz, and 1.9 GHz |
| Frequency Bands | Unlicensed | Licensed and unlicensed |
| Network Topology | Star, Mesh, and Hybrid | Star and Point-to-Point |
| Coverage Area | 10 km (rural), 2 km (urban) | 10 km (rural), 1 km (urban) |
| Battery Life | Up to 10 years | Up to 15 years |
| Data Rate | 0.3–50 kbps | 50–250 kbps |
| Security | AES-128 bit encryption | AES-128 bit encryption |
| Deployment | Requires a gateway | Cellular network required |
| Scalability | Can support thousands of nodes | Can support thousands of nodes |
| Latency | Seconds to minutes | Sub-seconds |
| Use Cases | Environmental monitoring, smart parking, asset tracking | Industrial automation, smart cities, security and surveillance |

One of the main advantages of LoRaWAN is its long-range communication capabilities, which make it suitable for use in cases that require devices to be deployed in remote areas, such as environmental monitoring or asset tracking. Additionally, the unlicensed frequency bands used by LoRaWAN make it a cost-effective solution, as no licensing fees are required. However, the trade-off for this long-range communication is a low data rate and higher latency, which may not be suitable for applications that require real-time data [24].

On the other hand, NB-IoT offers high data rates, low latency, and reliable connectivity in areas with weak signal strengths. Its cellular network infrastructure also provides a level of security and reliability that may not be possible with LoRaWAN. However, the licensing fees and higher deployment costs associated with the NB-IoT may make it less cost-effective than the LoRaWAN for certain applications [24].

In conclusion, both the LoRaWAN and NB-IoT have their advantages and limitations, and the choice between them will depend on the specific requirements of the application. The LoRaWAN is best suited for applications that require long-range communication and low data rates, while NB-IoT is ideal for applications that require real-time data and operate in areas with weak signal strength.

### 3.2. Assessing Financial Viability of Innovative Technologies

Life cycle costing (LCC) is a method for calculating the total cost of ownership of a product or service over its entire life cycle, from design and development to disposal. It is widely used in the field of advanced technologies, where the high initial cost and long life cycle of products require a comprehensive analysis of the total cost of ownership [25,26].

One of the key benefits of LCC is that it provides a comprehensive view of the costs associated with a product or service. This includes not only the initial purchase price but also the costs of maintenance, repair, and replacement over the life of the product. LCC also takes into account the impact of factors such as energy consumption, environmental impact, and regulatory compliance.

Terotechnology is a related concept that refers to the application of engineering and management principles to optimize the life cycle costs of physical assets. It is based on the idea that the cost of ownership of an asset is not just the initial purchase price, but also the cost of operating, maintaining, and disposing of the asset over its entire life cycle. Terotechnology considers the technical, economic, and social factors that affect the performance of an asset and seeks to optimize the cost-effectiveness of the asset throughout its life cycle [27].

While terotechnology has its merits, LCC is more pragmatic and has a better chance to be used in practice. This is because LCC is a more straightforward and easily understandable approach for calculating the total cost of ownership of a product or service. It is also more widely accepted and used in industry and government, with many organizations requiring LCC analyses as part of their procurement and purchasing processes.

One of the challenges of LCC is the need to gather accurate and reliable data on the costs associated with a product or service over its entire life cycle. This requires a detailed understanding of the product's design, manufacturing process, and operating characteristics, as well as the costs of maintenance, repair, and replacement over time. It also requires an understanding of the external factors that can affect the cost of ownership, such as changes in regulations, energy prices, and environmental policies [25,26].

To overcome these challenges, organizations can use a variety of tools and techniques to gather and analyze data on the life cycle costs of their products or services. These include cost accounting systems, enterprise resource planning (ERP) software, and specialized LCC software tools. These tools can help organizations to identify areas where costs can be reduced and to make more informed decisions about the design, development, and procurement of products and services [26].

Overall, LCC is a valuable approach for assessing the total cost of ownership of advanced technologies. By taking a comprehensive view of the costs associated with a product or service over its entire life cycle, LCC can help organizations to make more informed decisions about the design, development, and procurement of products and services. While terotechnology has its merits, LCC is more pragmatic and has a better chance to be used in practice [25].

### 3.3. Financial Viability of Selected IoT Communication Technologies

Life cycle costing is a crucial tool for making informed decisions about the economic feasibility of IoT communication technologies. IoT systems are typically composed of numerous devices with diverse functionalities and connectivity options, and estimating the total cost of ownership over the system's life cycle can be complex. Life cycle costing involves evaluating the costs of a system over its entire lifespan, from procurement and deployment to maintenance and disposal, taking into account all relevant cost components. By understanding the full cost profile of a technology, businesses can make more informed decisions about which IoT communication technologies are financially viable and sustainable in the long term. Financial viability is a critical factor in determining the feasibility and success of complex and future-oriented technologies, such as LoRaWAN and NB-IoT. These technologies, with their potential to revolutionize various industries through the Internet of Things (IoT), require a comprehensive evaluation of costs to make informed decisions regarding their adoption and implementation.

One-time costs, including the purchase of hardware and software, play a significant role in assessing the financial viability of these technologies. The initial investment required to acquire the necessary infrastructure, devices, and sensors can be substantial. It is crucial to consider the costs associated with procuring the hardware and software components,

as well as any additional customization or integration required for specific applications. Moreover, ongoing maintenance costs should be factored in, as these technologies often require regular updates, bug fixes, and security patches to ensure optimal performance and address emerging challenges.

Recurring costs are equally important to consider when assessing the financial viability of LoRaWAN and NB-IoT. Communication costs form a significant component, as these technologies rely on wireless connectivity to transmit data between devices and platforms. The expenses incurred by the data plans, network subscriptions, and infrastructure maintenance should be evaluated to determine the long-term financial implications of deploying and operating these technologies. Additionally, maintenance costs encompass not only routine maintenance, but also potential repairs or replacements of faulty components over the technology's lifespan. Properly accounting for these recurring expenses is essential for budgeting and ensuring sustained operations.

Operating costs, including energy costs, are another critical aspect of financial viability. LoRaWAN and NB-IoT technologies often involve numerous devices and sensors spread across a network, which consume power to function. The energy consumption associated with these technologies can be substantial, particularly in large-scale deployments. Assessing the energy requirements and estimating the associated costs are vital for understanding the ongoing operational expenses and optimizing energy efficiency.

In addition to the direct costs, the lifelong learning of employees must be considered. With the continuous advancement of technology, it is crucial to ensure that the workforce possesses the necessary skills and knowledge to effectively operate and maintain these complex systems. Investing in employee training, upskilling, and lifelong learning initiatives is crucial to keep pace with the evolving technological landscape. These costs, both in terms of time and resources, need to be factored into the financial evaluation of LoRaWAN and NB-IoT technologies.

It is important to note that the financial viability assessments for complex and future-oriented technologies should not be limited to individual cost components. The holistic evaluation of all costs, including one-time, recurring, and indirect costs, provides a comprehensive understanding of the long-term financial implications. By considering the complete cost spectrum, decision-makers can gain insights into the total cost of ownership and make informed choices regarding the adoption and implementation of these technologies.

As LoRaWAN and NB-IoT technologies continue to evolve and find applications in various sectors, understanding their financial viability is crucial for organizations and stakeholders. Robust financial analysis, encompassing both direct and indirect costs, helps in evaluating the return on investment, optimizing resource allocation, and mitigating financial risks. Moreover, considering the financial viability of these technologies provides valuable insights into their long-term sustainability and enables strategic decision-making for their successful implementation in the ever-changing technological landscape.

Several studies have addressed the life cycle costs of various IoT communication technologies, including LoRaWAN and NB-IoT [28]. A big number of research works conclude that, among the plethora of low-power wide area network (LPWAN) technologies, the cost-effectiveness of IoT is not certain for IoT service solutions.

Another study conducted by the authors in 2020 [29] compared the applicability, including the costs, of LoRaWAN and NB-IoT for industrial applications.

However, it is worth noting that these studies have some limitations. For example, they focused primarily on specific applications and did not consider the impact of the size and scale of the IoT system on life cycle costs. The following table highlights important aspects of the financial viability of NB-IoT and LoRaWAN technologies for IoT applications. Table 2 provides a general overview and comparison of the financial viability aspects between NB-IoT and LoRaWAN technologies for IoT applications. It is important to conduct a comprehensive analysis specific to the use case and context to obtain accurate financial viability assessments.

**Table 2.** Important aspects of the financial viability of NB-IoT and LoRaWAN technologies for IoT applications.

| Aspect | NB-IoT | LoRaWAN |
|---|---|---|
| Cost Structures | Higher initial equipment costs<br>Lower operating costs<br>Lower maintenance costs<br>Higher subscription fees | Lower initial equipment costs<br>Higher operating costs<br>Higher maintenance costs<br>Lower subscription fees |
| Revenue Potential | Limited revenue opportunities<br>Lower potential for direct revenue | Diverse revenue opportunities<br>Potential for direct and indirect revenue streams |
| Total Cost of Ownership | Relatively higher | Relatively lower |
| Cost Savings Opportunities | Potential for savings in equipment costs and subscription fees through economies of scale | Potential for savings in maintenance costs and subscription fees through |
| Decision-making Support | Requires careful evaluation of long-term operational costs and revenue potential long-term operational costs and | Requires consideration of application requirements, scalability, and specific needs |

To address these limitations, a holistic approach to life cycle costing is needed, one that takes into account not only the economic but also the environmental and social impacts of IoT communication technologies. While there are some studies that have applied life cycle costing to IoT systems in general, there is currently a lack of a holistic artifact that specifically addresses the economic and financial viability of LoRaWAN and NB-IoT technologies. Such an artifact would provide a comprehensive framework for evaluating the life cycle costs of these technologies, taking into account all relevant cost components. Additionally, it would allow for the comparison of the economic and financial viability of LoRaWAN and NB-IoT across a range of applications and scenarios.

## 4. Approach to Constructing the Scientific Artifact "Pragmatic Computational Tool" for Calculating the Life Cycle Costs of IoT Devices Based on Design Science Research

The present study aimed to develop a pragmatic computational tool using a design science research (DSR) approach for calculating the life cycle costs of IoT devices based on relevant parameters such as hardware (sensors and gateways), software costs, server costs, personnel-related costs, etc. The first step in the DSR approach was problem identification, which highlighted the lack of a comprehensive tool for life cycle cost analysis of IoT devices. The proposed tool aimed to fill this gap by providing a user-friendly and reliable way to calculate the life cycle costs of IoT devices that could be customized as per users' needs [30].

The design phase involved creating a model of the proposed artifact, which was a computational tool capable of taking various inputs, such as hardware, software, server, and personnel-related costs, and generating outputs, including the total cost of ownership, return on investment, and payback period. The tool was designed to be customizable, which enabled users to tailor the inputs and outputs to suit their specific needs [13].

The next step involved the implementation of the model in the form of a working prototype. The prototype was evaluated to ensure that it met the needs of the stakeholders, which included IoT device manufacturers, system integrators, and end-users. The prototype was evaluated based on its functionality, usability, and usefulness using methods such as user testing, expert reviews, and other forms of feedback [13].

Based on the feedback received, the prototype was refined and improved through an iterative process until it met the needs of the stakeholders. This iterative process of refinement and improvement is a hallmark of DSR. The final product was reliable, user-friendly, and met the needs of the stakeholders [13].

The development of the tool involved problem identification, model creation, implementation, evaluation, refinement, and communication of the results. The proposed tool

fills a significant research gap and provides a customizable, user-friendly, and reliable way to calculate the life cycle costs of IoT devices [13–15].

## 5. Constructing the Scientific Artifact "Pragmatic Computational Tool" for Calculating the Life Cycle Costs of IoT Devices

The "Pragmatic Computational Tool" for calculating the lifetime of IoT devices was built using Microsoft Excel and Google Sheets. These programs are commonly used tools for data analysis, financial modeling, and cost calculations. Decision-makers are likely to already have the necessary skills and familiarity, making the cost calculator more accessible and intuitive for them to use. By using a tool that decision-makers are already familiar with, the learning curve associated with adopting a new software or tool is minimized. The spreadsheet format lends itself well to organizing and structuring the various cost categories and tasks involved in calculating the lifecycle costs of IoT devices.

The tool was designed to provide a practical and user-friendly way to estimate the life cycle costs of IoT devices. The tool uses different categories of costs, including procurement costs, training and usage costs, maintenance costs, disposal costs, and external project costs, to estimate the total cost of ownership (TCO) of an IoT device over its lifetime.

To construct the tool, the first step was to create a worksheet in Excel with different categories of costs as column headers.

Formally, a section consists of a heading, a finer subdivision of the costs, fields for entries, and fields for the calculated costs (c.f. Figure 3).

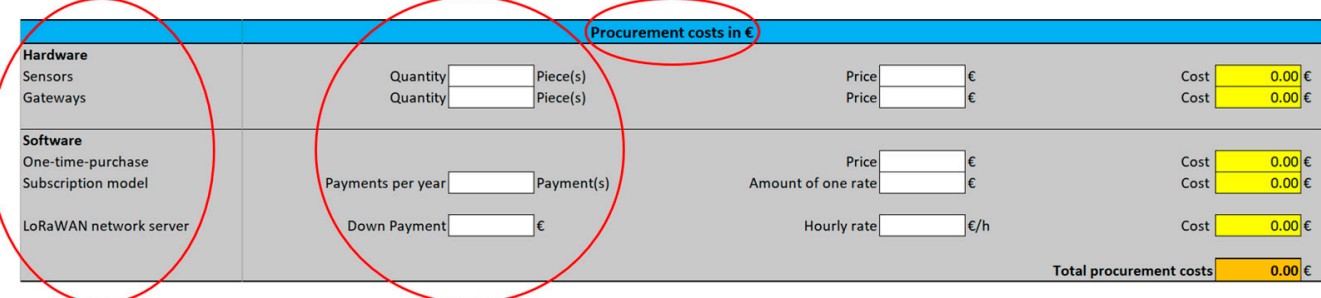

**Figure 3.** Design of the "Artifact" cost calculator; Own illustration.

The columns are labeled as procurement costs, training and usage costs, maintenance costs, disposal costs, and external project costs. The rows were labeled with specific tasks that are required to maintain and operate the IoT devices. For example, tasks such as hardware and software installation, training and support, device maintenance, disposal, and project management were included. Once the categories and tasks were identified, the next step was to assign cost values to each of them. The costs of a row are always summarized in a yellow field in the right column, and the cost of all lines in a section is displayed in an orange box (c.f. Figure 3).

Visually, the calculator is kept in unobtrusive gray, while the headings are highlighted in light blue. In addition, the color of individual fields varies depending on their meaning, ranging from to be filled in, via calculated automatically, to the sum above everything (c.f. Figure 4).

To make the tool even more user-friendly, symbols were used to represent the different types of costs. For example, a dollar sign (€) was used to indicate procurement costs, a wrench symbol was used to indicate maintenance costs, and a recycle symbol was used to indicate disposal costs (c.f. Figure 5).

- Fields for entries
  → white: „empty" „free".

- Fields that are automatically filled
  → green: „everything ok" „no action necessary".

- Fields for automatically calculated costs
  → yellow, orange: „signal color" „important".

- Field in which the total costs are shown
  → turquoise: „conspicuous" „stands out from the other cost fields".

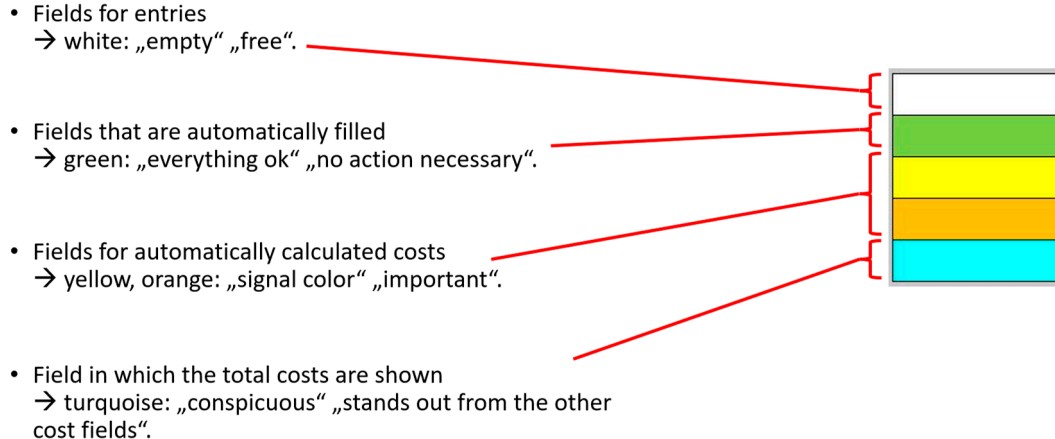

**Figure 4.** Color scheme of the "Artifact" cost calculator; Own illustration.

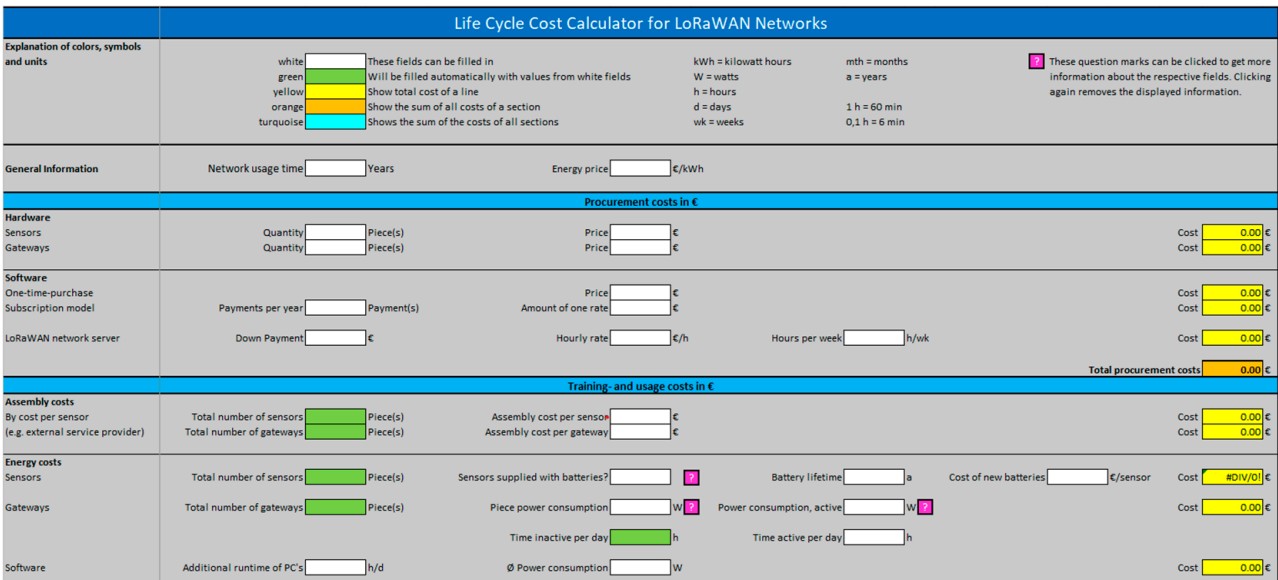

**Figure 5.** Interface of the Artifeact for the life cycle cost calculator for LoRaWAN; Own illustration.

In addition to assigning costs to each task, the tool also included units of measurement. This helped users understand the scale of the costs associated with each task. Once all the costs were assigned and the tool was complete, it was tested and validated to ensure its accuracy and usability. The tool was tested using different scenarios to determine its effectiveness in estimating the TCO of IoT devices. Feedback was collected from users to identify any areas of improvement, and the tool was updated accordingly.

Due to the different cost structures between IoT devices and gateways, it was necessary to construct separate calculators for each. As such, separate calculations are necessary to accurately estimate the total cost of ownership for each type of device (c.f. Figure 5).

Additionally, external project costs may also differ between IoT devices and gateways. For instance, the installation of gateways may require more specialized expertise and equipment, resulting in higher costs.

Overall, the "Pragmatic Computational Tool" provides a practical and user-friendly way to estimate the life cycle costs of IoT devices. It is easy to use, with well-explained colors, symbols, and units, making it an effective tool for decision-makers in the IoT industry.

## 6. Validating and Discussing the Scientific Artifact "Pragmatic Computational Tool" for Calculating the Life Cycle Costs of IoT Devices in a Smart City Environment

The evaluation and assessment of an artifact within the framework of design science research necessitate a systematic process, encompassing the establishment of clear evaluation goals, the design of the evaluation methodology, the collection and analysis of relevant data, the interpretation of results, and the reflection on findings to inform iterative improvements [13,14].

The validation of the scientific artifact "Pragmatic Computational Tool" for calculating the life cycle costs of IoT devices was conducted using several use cases from different domains, including smart cities, environmental monitoring, energy management, citizen science, and traffic management. The objective of the validation was to assess the accuracy and usability of the tool in various real-world scenarios and to identify any limitations or areas for improvement [31].

The assessment of the "Pragmatic Computational Tool" for calculating the life cycle costs of IoT devices was conducted using a validation process that involved several use cases from different domains. Out of the fifteen entities contacted, nine participated in the assessment, representing a diverse range of organizations employing either LoRaWAN or NB-IoT communication technologies in their smart city environments.

The assessment encompassed various smart city use cases representing distinct domains, including traffic management, environmental monitoring, citizen science, and energy management. The artifact was evaluated based on its ability to effectively monitor and optimize specific aspects within each domain. For example, in the traffic management domain, the artifact was assessed for its capability to monitor traffic flow, parking availability, and air quality. Similarly, in the environmental monitoring domain, the artifact's performance was evaluated in terms of monitoring air and water quality. The citizen science use case involved assessing the artifact's ability to collect data on weather conditions and noise levels. Lastly, the energy management use case focused on evaluating the artifact's effectiveness in optimizing energy consumption and production in buildings and industrial facilities. Through these assessments, this study aimed to determine the artifact's applicability and effectiveness in addressing the unique challenges and requirements of each domain within the context of smart cities.

In each use case, the tool was used to calculate the life cycle costs of the IoT devices, including procurement costs, training and usage costs, maintenance costs, disposal costs, and external project costs. The tool used different cost assumptions and parameters for each use case depending on the specific requirements and characteristics of the scenario.

The validation of the tool involved several steps, including verifying the accuracy of the calculations, assessing the usability and accessibility of the tool, and analyzing the results to identify any patterns or trends across the different use cases [32].

To verify the accuracy of the calculations, the tool was compared to other established methods for calculating life cycle costs, such as the traditional cost accounting approach and the Total Cost of Ownership (TCO) framework. The results showed that the tool was able to produce accurate and reliable cost estimates for each use case, and that the results were consistent with the results obtained from other methods.

To assess the usability and accessibility of the tool, the tool was evaluated by a group of experts in each use case domain. The experts were asked to evaluate the tool based on several criteria, including ease of use, clarity of instructions, and accessibility of the tool for non-experts. The feedback from the experts was positive, and they found the tool to be user-friendly and intuitive, with clear instructions and a simple interface. Table 3 delivers an overview on the assessment approach for the Pragmatic Computational Tool.

**Table 3.** Overview on the assessment approach for the Pragmatic Computational Tool.

| Assessment Activity Dimension. | Actual Activities |
|---|---|
| Use Case Selection | Identification of diverse domains: Smart city, environmental monitoring, energy management, citizen science, and traffic management. |
| Participant Engagement | Contacted 15 entities deploying LoRaWAN or NB-IoT communication technologies in smart city environments, of which 9 participated in the evaluation. |
| Assessment Objective | Evaluate usefulness, accuracy, usability, and identify areas for improvement. |
| Validation Steps | Qualitative interaction of results and analysis of results. |
| Accuracy Verification | Compare tool's calculations with traditional cost accounting and Total Cost of Ownership (TCO) framework. |
| Usability Evaluation | Expert evaluators assess ease of use, clarity of instructions, and accessibility for non-experts. |
| Analysis of Results | Identify patterns and trends across different use cases. |
| Findings and Recommendations | Extract insights and suggestions for tool enhancement based on the assessment outcomes. |

For the assessment, a total of 15 entities were initially contacted, representing diverse sectors and domains in the smart city context. Out of these fifteen entities, nine responded and actively participated in the assessment, providing valuable insights into their specific use cases. The assessment involved the utilization of the Pragmatic Computational Tool to calculate the life cycle costs of IoT devices in the domains of traffic management, environmental monitoring, citizen science, and energy management. In addition to using the tool, a qualitative interview approach was adopted to gather in-depth information and perspectives from the participating entities. This combination of quantitative analysis through the tool and qualitative interviews allowed for a comprehensive assessment of the artifacts' effectiveness and applicability in addressing the unique challenges and requirements of each use case within the smart city context.

The assessment results provide valuable insights into the costs associated with using LoRaWAN and NB-IoT technologies in various smart city use cases. The findings indicate that, initially, NB-IoT has lower costs compared to LoRaWAN for a smaller number of installed nodes (up to 5000 nodes). However, as the number of nodes increases beyond this threshold, the cost advantage shifts towards LoRaWAN, making it a more cost-effective option (c.f. Figure 6).

In the specific use cases assessed, it can be observed that for the Citizen Science domain employing LoRaWAN, the costs ranged from 8900 euros for 50 installed nodes to 20,800 euros for 500 nodes. Similarly, in the Traffic Management domain utilizing LoRaWAN, the costs ranged from 14,250 euros for 250 nodes to 100,000 euros for 1800 nodes. The Energy Management use case employing LoRaWAN had costs of 118,000 euros for 5000 nodes.

On the other hand, the Traffic Management use case employing NB-IoT had costs of 1200 euros for 50 nodes, while the Environmental Monitoring use case had costs of 5700 euros for 250 nodes. In the Energy Management use case utilizing NB-IoT, the costs were 119,000 euros for 5000 nodes.

These results indicate that NB-IoT technology initially offers cost advantages for smaller-scale deployments, as it can be integrated into existing cellular networks, resulting in lower hardware and installation costs. However, as the number of nodes increases, LoRaWAN becomes more cost-effective due to the use of lower-cost devices and the ability to support a higher number of nodes per gateway.

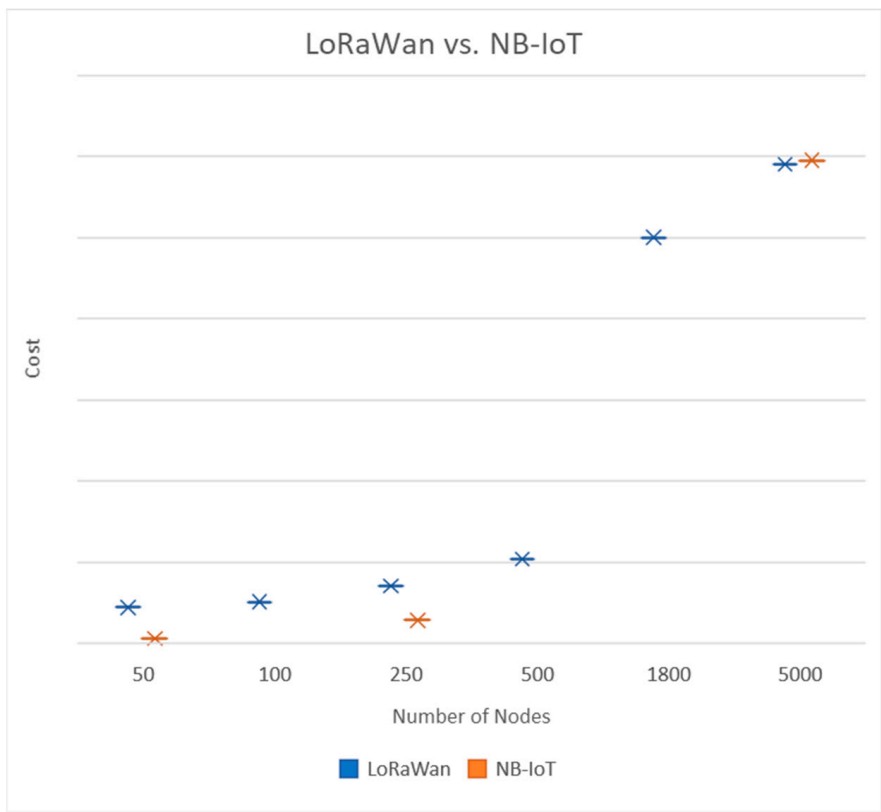

**Figure 6.** Cost of LoRaWan vs. NB-Iot.

These findings emphasize the importance of considering the scale of the deployment and the specific requirements of the use case when selecting the appropriate communication technology. Decision-makers need to carefully evaluate the cost-performance trade-offs between LoRaWAN and NB-IoT based on factors such as the number of nodes, hardware costs, installation expenses, and long-term maintenance requirements.

Overall, the assessment highlights the cost dynamics between LoRaWAN and NB-IoT, showcasing the advantages of each technology at different scales of deployment. These insights can inform decision-making processes for selecting the most cost-effective communication technology for IoT applications.

The cost performance disparity between the NB-IoT and LoRaWAN in scenarios with fewer sensors can be attributed to several factors. Firstly, LoRaWAN necessitates a workforce with specialized skills to operate and maintain the network, leading to higher labor costs. Additionally, the employment of gateways in LoRaWAN systems contributes to elevated hardware expenses per device. In contrast, NB-IoT offers advantages in terms of cost-efficiency by leveraging existing cellular networks, resulting in lower hardware and installation costs. However, as the number of devices escalates, LoRaWAN emerges as the more cost-effective option due to its utilization of economical devices and the ability to support a higher number of devices per gateway. This underscores the significance of conducting a meticulous analysis of the specific use case and requirements before determining the most suitable IoT communication technology.

Moreover, feedback received from stakeholders further supports the arguments. One recurring response was the preference for LoRaWAN, driven by the desire to have control over the technologies and data while ensuring future-proofing, especially when anticipating a large-scale deployment of nodes. Stakeholders with an IT-oriented perspective expressed enthusiasm for managing a new system, acknowledging that it could become a core competency for their organizations. Conversely, feedback from an NB-IoT use case highlighted the inclination towards NB-IoT to avoid the challenges associated with finding

qualified employees, suggesting that the ease of implementation and reduced dependence on specialized personnel was a significant advantage.

These additional arguments emphasize the multi-faceted considerations in selecting the appropriate IoT communication technology. Factors such as control over technologies and data, future scalability, management capabilities, and the availability of qualified employees all play critical roles in the decision-making process. Assessing the specific needs and aligning them with the strengths and limitations of each technology is vital for making informed choices and ensuring successful implementation in various use cases within the smart city context.

## 7. Assumptions and Limitations

This paper focuses on quantifying the economic and financial viability of NB-IoT and LoRaWAN technologies, which are two low-power wide-area network (LPWAN) technologies with unique characteristics suitable for IoT applications. This study aims to propose an artifact for performing life cycle cost analysis and demonstrate its application to these technologies. The methodology utilizes pragmatic computational tools to facilitate the analysis and considers various economic and financial factors, including operating costs, equipment costs, and revenue potential.

There are assumptions like the development of hardware and the communication costs for NB-IoT and LoRaWAN follow a similar trend over time. However, if the expected costs decrease, the evaluation may shift towards NB-IoT. The costs of employees actively involved in the LoRaWAN implementation are explicitly included.

The main finding of this study indicates that NB-IoT and LoRaWAN technologies have distinct cost structures and revenue potentials, influencing their economic and financial viability for different IoT applications. It concludes that a comprehensive life cycle cost analysis is crucial for informed decision-making regarding technology adoption. Furthermore, the proposed methodology can be applied to other IoT technologies to gain insights into their economic and financial viability.

In terms of future relevance, Design Science Research is particularly suitable for generating practical and impactful research outcomes while ensuring scientific rigor. This research approach will play an increasingly important role in the planning, design, and implementation of innovations and will gain recognition among research methodologies.

Further research is needed to validate the tool in other use cases and to refine the tool to better reflect the specific requirements and characteristics of each scenario. The cost calculators developed for NB-IoT and LoRaWAN aim to provide a quick and efficient way to estimate the expected costs of these networks. The calculators are designed to provide a clear overview of where and in which phase these costs are incurred, as well as to compare the two network types in terms of costs.

This study, however, has certain limitations. It is expected to yield incremental innovations rather than groundbreaking research results, with a focus on improving products, processes, and systems through validation. Additionally, the analysis did not consider other communication technologies. The evaluation of the artifact was conducted with a specific target audience involved in the Smart City domain. It is important to note that the calculators are not intended to evaluate the suitability of a particular technology and should not be used as a substitute for later cost accounting [33]. Additionally, the calculators cannot map all contingencies and special cases that may arise during the implementation and operation of these networks.

Furthermore, certain aspects are not considered in the cost calculators. These include the cost of capital, interest payments, depreciation, inflation/deflation rate, electricity price development, revenue generated by the network (in the case of LoRaWAN), safety aspects, network coverage, and other technical aspects of the networks. Therefore, it is important to use the calculators in conjunction with other tools and resources to fully evaluate the costs and suitability of each network type for a particular use case.

## 8. Conclusions

In this study, the validation of the "Pragmatic Computational Tool" for estimating the life cycle costs of IoT devices was carried out to assess its utility, reliability, and user-friendliness. The primary objective was to evaluate the economic and financial viability of Narrow Band-Internet of Things (NB-IoT) and Long Range Wide Area Network (Lo-RaWAN) technologies, both of which are low-power wide-area network (LPWAN) technologies specifically designed for IoT applications. To achieve this, a design science research approach was employed, which involved the development and subsequent evaluation of an artifact capable of performing a comprehensive life cycle cost analysis using practical computational tools.

The evaluation process was designed to provide valuable insights into the total cost of ownership associated with the utilization of NB-IoT and LoRaWAN technologies. By conducting a thorough assessment, this study aimed to identify potential cost-saving opportunities and facilitate informed decision-making processes regarding the adoption and implementation of these technologies in a range of IoT applications. Through this evaluation, the researchers sought to examine the distinct cost structures and revenue potentials associated with NB-IoT and LoRaWAN technologies across various IoT domains.

In the design science research paradigm, employing a limited number of use cases for evaluation is considered a valid approach [14]. This is primarily due to the emphasis on problem-solving and the iterative development process inherent in this research methodology. The focus is on creating innovative solutions and advancing knowledge in specific domains rather than conducting large-scale empirical studies. By analyzing a small number of carefully selected use cases, researchers can generate valuable insights and demonstrate the feasibility, effectiveness, and applicability of the developed artifact in addressing the identified problem.

While expanding the number of use cases and broadening the scope, such as including additional regions, may enhance the tool's acceptance and generate more comprehensive results, such an extension would exceed the scope of the present study. The current research aimed to provide a preliminary assessment of the Pragmatic Computational Tool within a limited number of use cases, showcasing its potential and addressing specific challenges within the examined domains. Further research could explore additional use cases to strengthen the generalizability of the findings and increase the tool's acceptance as a widely applicable cost calculator.

Furthermore, the proposed methodology, utilizing the Pragmatic Computational Tool, holds significant potential for application to other IoT technologies beyond NB-IoT and LoRaWAN. By adapting the tool to different IoT contexts, researchers can gain valuable insights into the economic viability of various technologies, enabling informed decision-making and resource allocation. The tool's flexibility and practicality make it a promising instrument for conducting comprehensive life cycle cost analyses in diverse IoT applications.

In conclusion, the validation of the "Pragmatic Computational Tool" provided favorable results, highlighting its utility, reliability, and user-friendliness in estimating the life cycle costs of IoT devices. The research demonstrated the effectiveness of employing a design science research approach, utilizing a small number of use cases to evaluate the artifact. While the study focused on NB-IoT and LoRaWAN technologies, it is acknowledged that expanding the scope and incorporating additional use cases would enhance the tool's acceptance and applicability. Moreover, the proposed methodology exhibits potential for application to other IoT technologies, thereby facilitating informed decision-making and resource allocation in various IoT domains.

**Author Contributions:** Conceptualization, B.K., M.B. and M.K.; methodology, B.K. and L.W.; software, M.K.; formal analysis, B.K. and L.S.; investigation, M.B., R.B. and M.W.; resources, B.K. and A.K.; writing—original draft preparation, B.K.; writing—review and editing, B.K., M.B. and T.B.; visualization, M.B.; funding acquisition, L.S. All authors have read and agreed to the published version of the manuscript.

**Funding:** This research received no external funding.

**Institutional Review Board Statement:** Not applicable.

**Informed Consent Statement:** Not applicable.

**Data Availability Statement:** Data is available at Pforzheim University upon request.

**Conflicts of Interest:** The authors declare no conflict of interest.

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
