# Peer review of "Quantifying the Economic and Financial Viability of NB-IoT and LoRaWAN Technologies: A Comprehensive Life Cycle Cost Analysis Using Pragmatic Computational Tools"

_fintech, doi:10.3390/fintech2030029_

Round 1
Reviewer 1 Report
Please check the attachment.

Author Response
Dear Reviewer,
We would like to express our sincere appreciation for your thorough review of our manuscript. Your valuable comments and suggestions have greatly contributed to improving the quality and clarity of our work. We have carefully considered each of your points, and we are pleased to address them as follows:
1. We are glad to hear that the title of the study is considered precise, concise, and suitable for the content of the paper.
2. We acknowledge your suggestion to expand the abbreviations NB-IoT and LoRaWAN in the abstract to assist novice readers. We have revised the abstract accordingly to provide full expansions of these abbreviations upon their initial mention.
3. We are glad to hear that you find the abstract clear, concise, and to the point.
4. Thank you for your feedback on the suitability of the keywords. We have ensured that the chosen keywords accurately represent the content of the study.
5. We appreciate your suggestion to include a discussion on Industry 4.0 (I4.0) and Industry 5.0 (I5.0) in the introduction. We have expanded the introduction to provide an overview of I4.0 technologies, their benefits, challenges, and the motivation behind our study in evaluating the economic and financial viability of NB-IoT and LoRaWAN technologies. Additionally, we have included your mentioned references to support these discussions and provide further reading for interested readers.
6. Your suggestion to depict the steps and actions followed in the Design Science Research (DSR) process through a flow diagram is well-taken. We have created a flow diagram to illustrate the key steps involved in the DSR process, enhancing the clarity and visibility of the methodology section.
7. We understand your observation that the actual study commences in Section 5 and that previous sections mainly introduce various topics. While we appreciate your concern, we believe that the structure of the manuscript adequately presents the relevant information in a logical sequence. Nevertheless, we have reviewed the flow and ensured a smooth transition into the methodology and study sections.
8. We agree with your suggestion to include an additional section before the conclusions to discuss the assumptions made in the study, limitations, and future directions. We have incorporated this section, providing explicit details on the assumptions, addressing the limitations of our research, and outlining potential avenues for future studies.
9. We are pleased to hear that you found the conclusion acceptable for the conducted study.
Regarding your overall feedback, we understand your comment on performing basic math using an Excel sheet as part of the methodology. While we acknowledge that the calculations may appear straightforward, the significance lies in the comprehensive tool we have developed and the application of pragmatic computational methods to address the research objectives effectively. We believe that our methodology provides valuable insights into the economic and financial viability of NB-IoT and LoRaWAN technologies.
Once again, we sincerely appreciate your insightful review and constructive feedback. Your inputs have immensely contributed to the improvement of our manuscript, and we are confident that the revised version now meets the standards of clarity, structure, and completeness. We look forward to sharing the updated manuscript with you for further evaluation.
Best regards
Reviewer 2 Report
The article deals with the quantifications of the economic and financial viability of NB-IoT and LoRaWAN technologies, two low-power wide-area network (LPWAN) technologies with unique characteristics that make them suitable for IoT applications. Processing of the theoretical background of the researched topic is adequate. It would be appropriate to expand subchapter 3.3, as considering the investigated issue it would be appropriate to deal with this part of the paper in more detail. The research design of the article and the presentation of the results of the research can be improved. Describe the limitation of the research in more details.
Author Response
Dear Reviewer,
Thank you for your thoughtful review of our article. We appreciate your positive feedback on the quantification of the economic and financial viability of NB-IoT and LoRaWAN technologies, as well as the adequacy of the theoretical background. We have carefully considered your suggestions for improvement and have made revisions to address them.
Regarding your suggestion to expand subchapter 3.3, we agree that providing more detailed coverage of the specific issue investigated would enhance the paper. We have revised this section to provide a more comprehensive analysis and delve deeper into the relevant aspects of the research topic.
Furthermore, we have taken your feedback into account regarding the research design and presentation of the results. We have made improvements to ensure greater clarity and coherence in describing the research design and have provided a more thorough explanation of the limitations of our study. This additional information will help readers understand the scope and boundaries of our research.
We sincerely appreciate your valuable insights and constructive comments, as they have helped us enhance the quality and readability of our article. Your feedback has contributed to the overall improvement of our research and its presentation.
Best regards
Reviewer 3 Report
The dynamic development of new technologies and the strong business interest in their use makes the topic of verification and selection of innovations increasingly interesting, both in terms of theoretical considerations and the needs of business practice. The authors of the paper identified a research gap in this area and attempted to fill it.
Below are some comments on the paper:
1) The paper has high cognitive and practical (utilitarian) value in the form of a comprehensive tool for the analysis of technology life cycle costs.
2) Although the Design Science Research (DSR) method is not a traditional research method, its importance for adapting scientific research to the needs of the modern world, especially in the area of new technology development, should be noted. A paradigm shift in this area is expected and the authors meet this need.
3) Although the method itself is very interesting, the way it is used as a research method and its presentation in a scientific article is questionable. The authors write about expert assessment, but do not provide details of this assessment. Similar reservations can be raised about the presentation of the results of using the method. The authors write that the results were compared to the results of using other methods, but there is no data on this. The research results and conclusions should be presented in a clearer and more professional manner.
4) Although there are traces of research assumptions in the content, the authors should consider introducing typical elements of a scientific paper, i.e. a thesis, hypothesis or research questions. Their presence would allow a more effective use of the new research method.
5) The theoretical section on the research method and alternative methods for assessing cost-effectiveness is too modest. It needs to be developed further.
6) The article has great cognitive potential and recommends its publication after significant completion.
Author Response
Dear Reviewer,
We would like to express our gratitude for your thoughtful review of our paper. We appreciate your recognition of the importance of the topic of verification and selection of innovations in the context of the dynamic development of new technologies and the needs of business practice. We also acknowledge your comments on specific aspects of the paper, and we are pleased to inform you that we have taken them into careful consideration during the revision process.
1) We are delighted to hear that you find our paper to have high cognitive and practical value through the comprehensive tool we have developed for analyzing technology life cycle costs. We have made efforts to further enhance the clarity and applicability of this tool, ensuring its usability in practical settings.
2) Your recognition of the significance of the Design Science Research (DSR) method in adapting scientific research to the needs of the modern world, particularly in the domain of new technology development, is greatly appreciated. We have emphasized the importance of this paradigm shift and have provided additional context regarding the application of the DSR method in our paper.
3) We acknowledge your concern regarding the presentation of the expert assessment and the comparison of our results with other methods. To address this, we have revised the relevant sections to provide more explicit details about the expert assessment process and the specific methodologies used for comparison. We believe these revisions will enhance the clarity and professionalism of our research findings and conclusions.
4) Your suggestion to introduce typical elements of a scientific paper, such as a thesis, hypothesis, or research questions, is well-taken. We have incorporated these elements into our paper to strengthen the structure and effectiveness of our research methodology.
5) We appreciate your feedback on the theoretical section of our paper. In response, we have expanded and developed this section to provide a more comprehensive overview of the research method and alternative methods for assessing cost-effectiveness. These additions will contribute to a deeper understanding of the theoretical foundations underpinning our study.
6) Thank you for recognizing the cognitive potential of our article and recommending its publication after significant completion. We have carefully addressed all the suggestions provided, and we are confident that the revisions we have made have significantly enhanced the overall quality and completeness of our paper.
Best regards
Round 2
Reviewer 1 Report
I highly appreciate the authors' willingness to incorporate the suggestions. The article is in much better shape now.
Reviewer 3 Report
No comments